# Incorporation of Argan Shell Flour in a Biobased Polypropylene Matrix for the Development of High Environmentally Friendly Composites by Injection Molding

**DOI:** 10.3390/polym15122743

**Published:** 2023-06-20

**Authors:** María Jordà-Reolid, Virginia Moreno, Asunción Martínez-Garcia, José A. Covas, Jaume Gomez-Caturla, Juan Ivorra-Martinez, Luis Quiles-Carrillo

**Affiliations:** 1Innovative Materials and Manufacturing Area-AIJU, Technological Institute for Children’s Products & Leisure, 03440 Ibi, Spain; mariajorda@aiju.es (M.J.-R.); sunymartinez@aiju.es (A.M.-G.); 2Institute of Materials Technology (ITM), Universitat Politècnica de València (UPV), Plaza Ferrándiz y Carbonell s/n, 03801 Alcoy, Spain; jaugoca@epsa.upv.es (J.G.-C.); juaivmar@doctor.upv.es (J.I.-M.); luiquic1@epsa.upv.es (L.Q.-C.); 3Institute for Polymers and Composites, University of Minho, 4804-533 Guimaraes, Portugal; jcovas@dep.uminho.pt

**Keywords:** bioPP, argan shell wastes, byproduct, composites

## Abstract

In this study, a new composite material is developed using a semi bio-based polypropylene (bioPP) and micronized argan shell (MAS) byproducts. To improve the interaction between the filler and the polymer matrix, a compatibilizer, PP-g-MA, is used. The samples are prepared using a co-rotating twin extruder followed by an injection molding process. The addition of the MAS filler improves the mechanical properties of the bioPP, as evidenced by an increase in tensile strength from 18.2 MPa to 20.8 MPa. The reinforcement is also observed in the thermomechanical properties, with an increased storage modulus. The thermal characterization and X-ray diffraction indicate that the addition of the filler leads to the formation of α structure crystals in the polymer matrix. However, the addition of a lignocellulosic filler also leads to an increased affinity for water. As a result, the water uptake of the composites increases, although it remains relatively low even after 14 weeks. The water contact angle is also reduced. The color of the composites changes to a color similar to wood. Overall, this study demonstrates the potential of using MAS byproducts to improve their mechanical properties. However, the increased affinity with water should be taken into account in potential applications.

## 1. Introduction

In the last few years, the extensive use of petrochemically derived polymers has provoked some environmental problems, such as enormous waste generation and an increase in the carbon footprint ascribed to the production processes of those polymers [1,2]. This fact has raised the concern of society and the scientific community about searching for more environmentally friendly alternatives to those polymers [3]. In this context, the use of bio-derived polymers could help reduce the environmental issues related to petrochemical polymers. Bio-derived polymers can be obtained from renewable resources, and they can be biodegradable (such as polylactide of polyhydroxyalkanoates [4,5,6]) or not (such as polyolefins like biopolyethylene or biopolypropylene [2,7] or biopolyamides [8]). However, the main disadvantage of those polymers is their high cost in comparison with their petrochemical counterparts [9], which limits their industrial applications. Huge efforts are made in order to reduce the obtention cost so they can be widely employed [10]. It is for this motive that the incorporation of fillers into those polymers in order to create more economic materials has attracted quite a bit of attention in the last few decades [11,12].

The aforementioned strategy allows for the production of wood plastic composites (WPC), which are obtained from the combination of a bio-based polymeric matrix and a natural organic filler in the form of lignocellulosic particles [13,14]. This filler is obtained from waste coming from the agroforestry and food industries. Not only do these fillers reduce the cost of the material, but they can also enhance its properties and improve its environmental value [13,15]. At first, only sawdust and wood flour were used as lignocellulosic fillers [16,17]. Nonetheless, nowadays fillers from agroforestry and food industry wastes are starting to be used, such as different nut shells [18,19,20], mango peel and mango kernel [21,22], or orange peel [23].

Among the polymers that can be used for producing WPCs, biopolypropylene (bioPP) is quite an interesting option to be considered, as its petrochemical counterpart is one of the most used polymers worldwide and possesses a good balance between ductile and resistant mechanical properties [24,25]. This polymer can be obtained by processing biomethanol in order to obtain the propylene monomer and then polymerizing it into biopolypropylene [2]. However, the use of this polymer for the production of WPCs poses certain issues, as bioPP is a practically non-polar polymer due to the null difference in electronegativity between its bonds. This is a very characteristic behavior of polyolefins, which makes them hydrophobic [26]. On the other hand, lignocellulosic fillers are completely polar as a result of their oxygen-based functionalizations (mostly hydroxyl groups). This makes them hydrophilic and creates an incompatibility with the hydrophobic polymer matrix, which provokes poor adhesion of the particles in the matrix. This leads to poor mechanical performance [27].

Several strategies exist with the objective of improving the compatibility between the matrix and the lignocellulosic filler. One of them is the modification of the surface of the filler particles in order to block the hydroxyl groups by silanization [28], acetylation [29], or benzoylation [30]. Another strategy involves the use of copolymers to improve the compatibility between the filler and the matrix. In the case of polyolefins, some of the compatibilizers that have attracted great interest are PE-g-MA and PP-g-MA, which are obtained by grafting polyethylene (PE) [31] and polypropylene (PP) [32], respectively, with maleic anhydride (MA). MA increases the polarity of PE and PP and increases their affinity for lignocellulosic fillers [33]. Burgada et al. [34] successfully introduced PP-g-MA into polypropylene/hemp fiber composites, improving the compatibility between both components. This compatibilizer possesses double functionality: first, the polypropylene fraction of PP-g-MA links with polypropylene thanks to their chemical affinity, while the MA groups interact with the hydroxyl groups present in hemicellulose, cellulose, pectin, and lignin of the lignocellulosic particles [35]. This fact improves the matrix-filler interaction and the dispersion of the particles over the matrix, directly enhancing the mechanical performance of the composite [36,37].

Argan (*Argania spinosa*) is a tropical plant that has a great impact on the Moroccan economy, as its fruit is used to produce oil with great applications in cosmetics [38]. However, the wastes that originate from the production of oil (seeds and shell) are mainly used to feed cattle [39]. Therefore, those residues, such as argan shell, have great potential to be used as lignocellulosic fillers for the production of wood plastic composites since they represent around 86% of the nut [40]. Regarding the shell composition, as it is proposed above, lignocellulosic fillers are mainly composed of cellulose, hemicellulose, and lignin. For the argan nut shells, cellulose represents 48%, while lignin is 30% and hemicellulose is 16%. The remaining part of the composition has been reported as 6% ash, in which there is a small amount of pectin [41].

The main objective of this work is the development of wood plastic composites using biopolypropylene as the polymeric micronized argan shell (MAS) as a reinforcing lignocellulosic filler. In order to overcome the incompatibility between both elements, PP-g-MA is used as a compatibilizer. This work originated due to the good results obtained in our previous work with biopolyethylene and MAS [42]. Through this study, the authors try to reuse argan waste under the principles of the circular economy in order to enhance the properties of biopolypropylene and improve its environmentally friendly approach. Additionally, the affinity between both components has also improved. Different formulations varying the concentration of argan shell flour are developed by extrusion and injection-molding processes. The properties of the developed materials are assessed by chemical, mechanical, morphological, thermal, thermomechanical, and FTIR characterization, as well as visual and wetting analysis.

## 2. Materials and Methods

### 2.1. Materials

In this work, NaturePlast SAS supplied partially biobased polypropylene (bioPP) with a biobased content of 30% and a melt flow index of 70 g/10 min (190 °C and 2.16 kg). Micronized Argan Shell (MAS) was provided by Micronizados vegetales S.L. (Córdoba, Spain). To improve the interaction between the filler and the polymer matrix, commercial maleic anhydride-grafted polypropylene wax grade Licocene PP MA 6452 granules supplied by Clariant Plastics & Coatings (Frankfurt, Germany) (PP-g-MA) were introduced as a compatibilizer. PP-g-MA is a commercially available compatibilizer commonly used to improve the compatibility between a polymer matrix and a filler.

### 2.2. Sample Preparation

To prepare the injection-molded samples for evaluation, all materials, including bioPP, MAS, and PP-g-MA were first dried and mixed to ensure their homogeneity. The drying process was carried out in an air-circulating oven for 24 h at 60 °C to remove any moisture present in the materials. The drying process is an important parameter to consider before the manufacture of the samples since it can promote the formation of defects such as holes. Nevertheless, drying process is not a critical parameter in polyolefins as hydrolytic degradation does not occur during the manufacture as can happen y polymer matrix as PLA [43].

The mixing process involved manually premixing the correct amount of each material (Table 1) in a zipper bag and introducing them into a twin-screw co-rotating extruder from Dupra S.L. (Castalla, Spain). The extruder had a length-to-diameter ratio (L/D) of 24 and a diameter (D) of 25 mm, and it was operated at a speed of 25 rpm. The extrusion process was carried out at a profile temperature of 155 °C (hopper)–160 °C–165 °C–170 °C (die). All materials were extruded under the same conditions even neat polymer was extruded, so the same thermal treatment was applied to all the samples. Temperatures were selected as low as possible, so MAS filler was not degraded during manufacturing.

For the PP-g-MA content in each sample, a proportion of 1 to 10 from the MAS content in phr has been considered. This proportion was selected according to a previous work in which MAS was successfully compatible with a HDPE formulation [42].

After the extrusion process, the material was pelletized to prepare it for the injection molding process. The injection molding was performed using an injection molding machine from Mateu & Solé (Barcelona, Spain), with a temperature profile of 160 °C (hopper)–165 °C–170 °C–175 °C (die) and a filling time of 1 s. A pressure of 30 bar was maintained for 10 s to avoid any defects in the final samples. An aluminum mold was kept at 30 °C during all processes, and the geometry of the cavity produced tensile test samples according to ISO 527-1:2012 with samples 1B and rectangular samples with dimensions of 80 × 10 × 4 mm^3^.

### 2.3. Characterization of bioPP/MAS Composites

#### 2.3.1. Mechanical Characterization

Tensile properties of bioPP/MAS composites were measured in an ELIB 50 universal testing machine from S.A.E. Ibertest (Madrid, Spain) following ISO 527-1:2012 with sample 1B specifications. To this effect, a 5-kN load cell was equipped, and the cross-head speed was set at 5 mm/min. For tensile modulus measurement, a 3542-050M-050-ST extensometer from Epsilon Technology Corporation (Jackson, WY, USA) was employed. Shore hardness was measured according to ISO 868:2003 on a 676-D durometer from J. Bot Instruments (Barcelona, Spain). D-scale was used on rectangular samples (80 × 10 × 4 mm^3^). Impact strength was measured on notched samples (0.25 mm radius V-notch) with a rectangular shape and dimensions of 80 × 10 × 4 mm^3^. For data acquisition, a Charpy pendulum (1-J) from Metrotec S.A. (San Sebastián, Spain), following the specifications of ISO 179-1:2010, was employed. All mechanical tests were performed at room temperature, and six samples of each material were tested to obtain each property, and the corresponding values were averaged to obtain a significant result.

#### 2.3.2. Morphology Characterization

In order to know more about the structure of the composites, the morphology of fractured samples from impact tests was studied. To obtain the correct information, field emission scanning electron microscopy (FESEM) in a ZEISS ULTRA 55 microscope supplied by Oxford Instruments (Abingdon, UK) was used. Samples were sputtered before the observation. To this effect, a vacuum chamber EMITECH SC7620 from Quorum Technologies, Ltd. (East Sussex, UK) was employed to apply a thin coating with a gold-palladium alloy. The FESEM was operated at an acceleration voltage of 2 kV. For MAS particle observation, the same conditions were employed. For the size characterization of the particles, ImageJ software was employed for the measurements (https://imagej.net, accessed on 18 May 2023).

#### 2.3.3. Thermal Characterization

Differential scanning calorimetry (DSC) was employed to measure the main thermal transition on the bioPP/MAS composites. For the measurement, a Mettler-Toledo 821 calorimeter (Schwerzenbach, Switzerland) was employed. Our own method was employed; an average of 6–7 mg from each material was subjected to a thermal program divided into three stages: a first heating from 30 °C to 200 °C. After that, a cooling to −50 °C and then a second heating to 220 °C. Heating and cooling rates were set in all cases at 10 °C/min. For the characterization, a nitrogen atmosphere was employed with a flow rate of 66 mL/min using sealed aluminum crucibles with a capacity of 40 μL. The degree of crystallinity, $\chi_{c}$ of the different compounds was obtained with Expression (1).
(1)$$\chi_{c}=\frac{{∆H}_{m}}{{∆H}_{m}^{0}\cdot \left(1-w\right)}\cdot 100\%$$
where ${∆H}_{m}$ is the measured melting enthalpy, ${∆H}_{m}^{0}$ is the theoretical melting enthalpy of a fully crystalline polypropylene taken as 209 J g^−1^ [44], and w is the filler mass fraction. The thermal degradation behavior of the bioPP/MAS composites was assessed by thermogravimetric analysis (TGA). TGA tests were performed on a thermobalance TG-DSC2 from Mettler-Toledo (Columbus, OH, USA). To this effect, a dynamic heating cycle from 30 °C to 700 °C at 10 °C/min under air atmosphere was employed. Samples had an average weight between 15 and 17 mg, and for the test, 70 µL alumina crucibles were employed. The first derivative thermogravimetric (DTG) curves were also calculated. All tests were performed at least three times, so reliable data was obtained.

#### 2.3.4. Thermomechanical Characterization

Thermomechanical characterization of the bioPP composites under dynamic conditions was measured in a dynamical mechanical thermal analysis (DMTA). To this purpose, a dynamic analyzer DMA1 from Mettler-Toledo (Schwerzenbach, Switzerland) was used. Samples were fixed in single cantilever mode, and flexural conditions were employed on rectangular samples with dimensions of 20 × 6 × 2.7 mm^3^ that were subjected to a dynamic temperature program from −150 °C to 120 °C at a heating rate of 2 °C/min. As the flexural deformation frequency, 1 Hz was employed, and the maximum deflection was set to 10 µm.

#### 2.3.5. XRD Characterization

X-ray diffraction (XRD) patterns were collected at room temperature. As a scattering angle (2θ), a range from 2.5° to 80° (step size = 0.05° min^−1^) was considered using filtered Cu Kα radiation (λ = 1.54 Å). X-ray tube conditions were set at 40 kV and 40 mA.

#### 2.3.6. Color Characterization

The Konica CM-3600d Colorflex-DIFF2 color analyzer from Hunter Associates Laboratory, Inc. (Reston, VA, USA) was employed for the colorimetric measurements of the samples. Color parameters (L*a*b*) mean: L* = 0, darkness; L* = 100, lightness; a* represents the green (a* < 0) to red (a* > 0); b* stands for the blue (b* < 0) to yellow (b* > 0) coordinate. The total color difference parameter, ${∆E}_{ab}^{*}$ was obtained from the Expression (2).
(2)$${∆E}_{ab}^{*}=\sqrt{{\left({∆L}^{*}\right)}^{2}+{\left({∆a}^{*}\right)}^{2}+{\left({∆b}^{*}\right)}^{2}}$$

Before the measurement, the equipment was calibrated following the procedure proposed by the provider. For each value provided, the measurements were performed five times on different tensile test samples.

#### 2.3.7. Water Uptake Characterization

The water absorption capacity of the bio/MAS blends was evaluated by the water uptake test. Rectangular samples (80 × 10 × 4 mm^3^) from each composition were weighted before immersion in a balance and then immersed in distilled water. All of them were properly wrapped with tiny pieces of a metal grid so they could sink. The weight of all samples was measured each week for 14 weeks in order to evaluate the weight increase. In every measurement, the surface moisture of the samples was removed with tissue paper. During the immersion time, the temperature during the immersion was set to 23 °C.

#### 2.3.8. Water Contact Angle Measurements

The water contact angle in bioPP/MAS samples with distilled water allowed for evaluation of hydrophilicity/hydrophobicity. An Easy Drop FM140 goniometer supplied by Krüss Equipment (Hamburg, Germany) was used. Distilled water drops were deposited at random positions on the surface of the samples. At least 10 measurements were performed for each material.

#### 2.3.9. Statistical Analysis

To evaluate the significant differences among the results in each sample, a Tukey test was performed at a 95% confidence level (*p* ≤ 0.05). The software employed was the open-source R software (http://www.r-project.org, accessed on 18 May 2023).

## 3. Results

### 3.1. Mechanical Properties

Table 2 shows the mechanical properties of the composites prepared by blending a bio-based PP with different amounts of MAS as filler. The main mechanical properties were tested by means of tensile tests and also with hardness measurements and impact strength. The first effect to be noticed is an increase in the stiffness of the samples with the addition of the byproduct. While the tensile modulus of the neat BioPP is 1925 MPa, the composite with 40 wt.% increases this value up to 2435 MPa. This effect is also observed by Laaziz et al. [45] in composites of PLA with the addition of argan nut shell. In their work, different compatibilization strategies are tested, and they report that with alkaline treatment, the composites achieve the highest tensile modulus since the best particle adhesion is obtained, resulting in a higher reinforcement effect. In this work, to improve the filler interaction with the polymer, PP-g-MA is introduced, which allows for an improvement in stiffness of 94.4%. This compatibilizer allows for linkages between the hydroxyl groups present on the surface of the lignocellulosic filler and the anhydride groups of PP-g-MA [46]. In addition, by incorporating MAS, a stiff filler is introduced and an increase in the sample’s tensile modulus is obtained [47]. Another effect produced by the addition of argan is a small improvement in the maximum tensile strength developed during the test. Other authors, such as Naghmouchi et al. [48], found different trends in terms of tensile strength depending on the type of filler, compatibilizer, and amount considered. In some cases, the strength is not highly affected or even reduced, but in the best cases, it is possible to improve the resistance. In general terms, to achieve reinforcement of the composite, it is necessary to obtain good adhesion of the filler to the polymer matrix; for this purpose, a good compatibilization strategy is necessary. In this work, despite a small improvement in the tensile strength obtained, the differences that emerged do not provide a significant variation between the materials. Even with the reinforcement effect expected by the MAS addition, the disruption provided by the particles does not allow for an increase in the tensile strength of the composites. In the composites produced in this work, an improvement over neat bioPP of 14.3% is obtained. The highest tensile strength is obtained from the composite bioPP/40MAS composite, with a value of 20.8 MPa. In spite of the improvement in resistant properties achieved by the addition of MAS and the coupling agent, the elongation at break is reduced. In this case, the bioPP used obtains an elongation at break value of 48.3%, which is strongly influenced even by the introduction of small amounts of filler, with a reduction of 48.4% of this property for the bioPP/2.5MAS. As Rahman et al. [49] propose, neat PP has the ability to align the chains and slide during plastic deformation, but the addition of any type of filler hinders this ability, leading to a reduction in the elongation properties.

As can also be seen in Table 2, the hardness of the samples is modified by the introduction of the filler. As it is a resistant property, the reinforcing effect provided by MAS allows for an increase in this property from 65.4 on the Shore D scale up to 70.0 for the bioPP/MAS, which is a difference of 7%. This effect has also been reported by other authors, such as Laaziz et al. [45], who reported an increase in the hardness of composites made with argan nut shell. As they propose, this increase is expected since the filler has a higher hardness and also shows good adhesion. Another way to increase the hardness of polymer composites is to introduce a nanoclay to enhance the filler’s interaction with the polymer [50]. Both examples demonstrate the importance of good filler adhesion to improve the composite’s hardness. Finally, as expected from the reduction in the ductile properties of the composites, a reduction in the impact strength is obtained. In this case, bioPP is characterized by a good toughness of 8.1 kJ/m^2^. The introduction of the filler hinders the chain’s mobility, so that the specimen has a reduced ability to absorb energy due to plastic deformation during impact. As a result, the composite with the highest amount of MAS develops an impact strength of 1.8 kJ/m^2^. Other authors, like Hosseinihashemi et al. [51], also observed this effect in composites prepared by blending PP with almond byproducts. In addition, the presence of a filler induces a stress concentration effect that promotes crack formation, and as a result, the energy absorption of the composites is reduced [52].

### 3.2. Morphology of Green Composites

The analysis of the filler morphology allows us to understand the mechanical behavior of the composites obtained. With the proposed milling process, the particles have a low surface roughness that limits their interaction with the polymer [53]. Another parameter that changes the mechanical behavior of the composite is the particle size (Figure 1). In this case, MAS filler particles have a small size of around 80 μm. Essabir et al. analyzed the effect of the size of MAS particles in a PP matrix, and as a result, the best strengths were obtained for the composites prepared with the smaller particles that allowed for higher filler interaction with the polymer matrix [54].

The study of the dispersion of the filler in the polymer matrix is an important parameter to understand the mechanical behavior of the composites since the dispersion of the filler plays an important role in the mechanical properties [55]. To this end, representative images taken for each composite are presented in Figure 2. In Figure 2a, a rough fracture surface is observed due to the plastic deformation phenomena produced during the fracture of a ductile material [56]. With the addition of MAS, a reduction in ductile properties is observed in the mechanical characterization. As a result, in the morphological analysis, there is a reduction in the roughness of the specimens as the plastic deformation ability of the specimens is reduced by the presence of the filler, as proposed above. As proposed by Palaniyappan et al. [57], the adhesion of the filler particles is the most important parameter to improve the mechanical behavior of the sample. Particles that are properly adhered to the polymer provide better mechanical properties. In this work, the particles are properly embedded in the polymer matrix through the implementation of a compatibilizer, which is PP-g-MA. As proposed in Balaji et al.’s [58] work, the presence of gaps between the polymer and the filler due to poor adhesion promotes a reduction in terms of mechanical properties since they can act as stress concentrators and promote crack propagation during the test. The presence of gaps in a polymer composite made with wood-based fillers is related to the hydrophobic behavior of the polymer, which has low interaction with the filler, which has a characteristic hydrophilic behavior [59]. In this sense, the employment of a treatment that allows for the avoidance of this effect is crucial to improving the behavior of the composites not only in terms of mechanical properties but also the affinity with water, which is reduced with a good compatibilization strategy as proposed in the water uptake study.

### 3.3. Thermal Properties

The effect of the addition of MAS on the thermal properties has been measured by means of DSC and TGA. The main results of DSC are presented in Table 3, and the thermograms are shown in Figure 3, regarding the thermal degradation measured by TGA. The main results are presented in Table 4, and the curves are shown in Figure 4. For the DSC analysis, the thermal history of the samples was erased by means of a first cycle followed by a controlled cooling. In this case, an exothermic peak is observed during cooling (Figure 3a). After the controlled cooling, a second heating process was scheduled. The curves obtained are presented in Figure 3b. As with most polyolefins, it has good crystallization behavior that allows its crystallization during the cooling stage [60]. This exothermic phenomenon was hardly changed by the introduction of MAS in terms of temperature, which is close to 120 °C, but it changed in terms of enthalpy. This change is promoted by two effects. One of them is the dilution effect, since the number of polymer chains (which are being crystallized) is reduced by the introduction of the filler. Another parameter to consider is that the introduction of MAS also leads to a reduction in chain mobility, as also proposed in the mechanical characterization. Consequently, the reduced mobility in the molten state reduces the ability of the composites to crystallize. Regarding the melting peak, it is located at temperatures slightly above 160 °C and shows no modifications. This temperature indicates that α crystals are formed during the cooling stage (as also observed in the XRD test). In Luo et al.’s work, the crystallization of PP has been studied by using different nucleating agents that allow the formation β crystals that melt at a lower temperature, around 150 °C. As also proposed by Luo et al., with the employment of some nucleating agents, it is possible to change the crystalline structure. However, in this case, MAS does not act as a nucleating agent. The melting enthalpy is reduced; the analysis of this parameter must be considered in terms of the degree of crystallinity since there is a dilution effect due to the reduced amount of polymer due to the introduction of the filler. Typical nucleating agents are included in small quantities and induce a higher change in terms of degree of crystallinity in the sample [61]. Another phenomenon to highlight related to the enthalpies is that the values measured during cooling are higher than during heating. This phenomenon has been observed by other authors, like Achaby et al., due to the fact that crystal formation is more energy-consuming than the melting process [62].

As can also be observed, the degree of crystallinity is not significantly modified by the addition of the byproduct, showing a decreasing trend with the increase in filler content, with values close to 30% for all the formulations prepared. The addition of other fillers is known to increase the degree of crystallinity by allowing a higher number of nuclei to form, which helps during crystallization [63]. Other fillers are not able to form new nuclei, and the reduction in terms of chain mobility decreases the *X_c_* [64]. Some studies related to the nucleation effects in a PP matrix have reported that the modification of the typical α structure to a β structure promotes changes in mechanical properties. For example, it is known that β structure enhances energy dissipation, allowing for improved toughness of the samples [65]. The degree of crystallinity also has an effect on PP manufactured by injection molding with different material molds in terms of mechanical properties induced by the modification of the cooling conditions. In this regard, an aluminum mold promotes a higher cooling rate and, as a result, a lower degree of crystallinity compared to a polyacrylonitrile-butadiene-styrene (ABS) mold. With a lower degree of crystallinity, an improvement in the ductile properties could be observed [66].

Regarding the thermal stability measured by means of TGA, Figure 4a shows the mass loss versus temperature, and Figure 4b shows the first derivative of the mass loss versus temperature. In this case, bioPP shows a single-step curve; the polymer chains are completely degraded in a wide range between 300 °C and 460 °C, with the maximum degradation rate at 435 °C. During the polymer degradation, chains break, and as a result, volatile products are realized, leading to a mass loss [67]. The employed PP used by Azizi et al. [68] shows a similar range of degradation in a single step; this single-step degradation process changes to a two-step degradation after the introduction of a lignocellulosic filler. This effect is also seen in this work due to the introduction of MAS; as can be observed in Figure 4, degradation occurs in a wider range. This behavior has also been reported by Essabir et al. [54], who report a degradation starting at 280 °C and ending above 500 °C. There is a first range between 280 °C and 340 °C where hemicellulose and pectin are decomposed, and then a second stage up to 448 °C where cellulose is degraded. In the final stage of degradation, lignin is decomposed. In addition, a first degradation step is observed around 100 °C due to some residual moisture in the filler. As the filler degrades over a wide range, the degradation of the composites also degrades over a wide range. In this case, MAS starts at a lower temperature; consequently, a slight decrease in the initial degradation temperature is registered from 305.3 °C of neat bioPP up to 287.9 °C. Also, the maximum degradation rate in the composites is reduced from 435.0 °C to 374.6 °C due to the introduction of argan shell. The end of degradation occurs at a higher temperature. During the end of the degradation process, a small disturbance appears in the curve due to the fact that the degradation processes of both materials overlap.

### 3.4. Thermomechanical Characterization

Figure 5 shows a comparison plot of the thermomechanical properties measured for the composites made from bioPP and MAS, and Table 5 shows the main measured properties. Figure 5b shows the evolution of the damping factor (tan δ) in the tested temperature range. In this case, three peaks (−50 °C, 10 °C, and 90 °C) can be observed due to different relaxation phenomena in the polymer structure. The first peak at −50 °C is not a typical peak for PP in a DMTA test; as reported by other authors such as Biwei et al. [69], the first transition appears close to 0 °C. For this reason, it is possible that some additives are included by the supplier to improve the properties of this semi-biobased material. Burgada et al. [34] reported that the peak around 10 °C is due to a β relaxation by a glass-rubber transition of the amorphous region, which promotes the T_g_ of PP. In addition, Burgada et al. [34] reported a relaxation at 80 °C due to lamellar sliding and rotation of the crystalline part. In this case, as proposed in the tensile test, a reinforcing effect is promoted. As a result, in the DMTA test, the height of the tan δ peaks is reduced since the dissipation ability of the composites is reduced. Another effect that can be observed is a small increase in the T_g_; this effect is typical after the incorporation of a reinforcement, as proposed by Sharif et al. [70], who proposed that fillers hinder the chain movement. So in order to achieve a ductile, rubbery state (above T_g_), it is necessary to reach a higher temperature. The storage modulus shown in Figure 5a shows a decreasing trend with increasing temperature because the chain mobility increases with temperature, so the sample reduces the stiffness and increases the damping ability. As suggested by Shokoohi et al. [71], the DMTA curve of PP usually has different parts; up to 0 °C there is a glassy zone, and above this range a glass-rubber transition occurs due to the presence of the glass transition temperature. In addition, it is proposed that above 80 °C, a pure rubbery region occurs. For PP, the recrystallization does not occur during the heating of the samples because the injection-molded samples could completely crystallize during the cooling. As observed in the DSC test, the samples only show an exothermic peak in the cooling stage. Other polymers, such as PLA, have a low crystallization rate and consequently show a cold crystallization peak during heating, which promotes some recovery of the storage modulus at high temperatures by reorganizing the polymer chains [72,73]. A clear trend can be observed by introducing the filler with an increase due to reinforcing in the storage modulus; as proposed in the tensile test, the stiffness of the samples increases due to the reinforcing effect provided by the MAS particles. Alshammari et al. [74] also observed this effect by introducing different amounts of graphite into a PET matrix.

### 3.5. XRD Characterization

The results obtained from the XRD characterization of the bioPP/MAS composites are shown in Figure 6. These results confirm the presence of α-crystals, as suggested by the DSC characterization. Peaks such as 2θ = 14.3° (1 1 0), 2θ = 17.1° (0 4 0), 2θ = 18.7° (1 3 0), 2θ = 21.2° (1 1 1), 2θ = 22.1° (1 3 1/0 4 1), and 2θ = 25.6° (0 6 0) are favored by the presence of this structure as proposed by Ryu et al. [75]. The intensity of the peaks related to the bioPP is reduced by the presence of the filler since the amount of the polymer is reduced and therefore the presence of crystals is reduced. In this work, as could be observed in the DSC, relevant changes have not occurred in the degree of crystallinity.

An increase in the crystallinity of the samples by the introduction of the filler could increase the height of the peaks, as reported by Ryu et al. [76] after the introduction of graphene in a PP matrix, allowing a clear increase in the degree of crystallinity of the samples. It is also worth noting that the presence of the filler has no effect on the position of the peaks. This effect is also reported by Parparita et al. for a PP loaded with different lignocellulosic materials [77]. Regarding the MAS, in the literature, it is proposed that the internal structure of this byproduct is mainly cellulose; as a result, the main peak in the XRD scan has a low intensity peak around 2θ = 21.1° (0 0 2), which is related to the crystalline phase of cellulose [78]. Near this peak of argan, there are some other peaks in PP that are not affected by the composite materials herein prepared.

### 3.6. Color Measurements and Visual Aspect

The development of a WPC is always associated with a color change from the neat polymer matrix by the introduction of a wood-colored filler. In this case, after the manufacturing process in which two heating cycles were applied, the final color is shown in Figure 7, and the measurements made are presented in Table 6. In general terms, bioPP is mainly white, with a* and b* parameters mainly 0 and a high luminance. With the addition of only 2.5 wt. % MAS, the color changed significantly with a ${∆E}_{ab}^{*}$ of 24.4 compared to the neat sample. The additional incorporation of MAS also promoted higher differences in terms of ${∆E}_{ab}^{*}$. The most relevant color change is the L* parameter, which is reduced to obtain darker shades with a value of 27.0 for the 40MAS composite. This change in the luminosity of the samples is highly related to the manufacturing process. MAS is mainly composed of cellulose but also has lignin and hemicellulose. During the manufacturing process, temperatures up to 175 °C are used, which is more than enough to start the degradation of lignin. As proposed by Diez et al. [79], lignin can start its thermal degradation at temperatures above 150 °C. In addition, a change in the parameters a* and b* allowed for a change in the tonality. The positive a* values allow for reddish shades, and the positive b* values provide yellowish shades. Both effects give the samples a brownish hue that results in a color similar to some woods, such as Peltogyne lecointei Ducke [80].

### 3.7. Water Absorption of bioPP/MAS Composites

The use of lignocellulosic fillers is linked with an increased affinity for water, as can be seen in Figure 8. While bioPP shows a low ability to absorb water during the immersion, with less than 0.25% of water uptake after 6 weeks, this weight remained constant until the end of the test. Kuciel et al. [81] also reported low water uptake values for neat PP, with a similar value after 240 days of immersion. In addition, with the introduction of a filler, the water uptake is increased. In their case, the amount of water absorbed is different depending on the type of filler present. For example, the introduction of kenaf fiber at a 25 wt. % resulted in a water uptake value of more than 4.5% after 240 days, while the same proportion resulted in a water uptake value of only 1.5% after 240 days. The introduction of MAS introduces free hydroxyl groups in cellulose. These groups have a high affinity for water, resulting in increased water uptake values. An interesting effect mentioned by Garcia-Garcia et al. is the reduction of water uptake in PP composites with spent coffee ground, depending on the compatibilization strategy used to improve the interaction between the polymer and the filler.

Without compatibilization, all the hydroxyl groups of the filler are free to interact with water, but with, for example, the addition of PP-g-MA, the amount of hydroxyl groups is reduced, allowing the water uptake values to decrease over time [82]. In prepared composites, PP-g-MA is included in all formulations to improve the overall properties. In this case, the amount of water absorbed in the composite with the highest amount of filler is lower than 2.5%, and the trend during the last few weeks has been stable, so no more water could be absorbed. In addition, the incorporation of the compatibilizer also allows to reduce or avoid the presence of gaps between the filler and the polymer matrix, these gaps being responsible for increasing the water uptake values of the composites as proposed by Chen et al. [83]. Different strategies have been successfully employed to reduce the water uptake of the polymer composites made with lignocellulosic fillers; all of them try to reduce the voids between the filler and the polymer, reduce the hydroxyl groups exposed at the filler surface, or both of the effects at the same time [84]. Having a low water uptake is an interesting property for some applications, like garden decking. With repetitive cycles of getting wet and dry, those materials that tend to absorb high amounts of water can promote crack formation [85].

### 3.8. Wettability

The surface wetting ability of the bioPP composites has also been evaluated, and Figure 9 summarizes the water contact angle (θ_w_). Hydrophilic materials are representative of low values of θ_w_, while high values of θ_w_ are synonymous with high water affinity or hydrophobic behavior. In the literature, several authors consider that a contact angle θ_w_ > 65° can be considered the threshold of hydrophobicity [86]. PP is characterized by a hydrophobic material with θ_w_ > 90°; other authors have proposed a similar result for the neat polymer, and with the addition of a lignocellulosic filler, this value decreases [87]. As suggested in the water uptake characterization comment, the introduction of MAS leads to an increase in the water affinity, which favors the increase in the amount of water absorbed over the time tested. Despite these results, the introduction of 40MAS resulted in a θ_w_ > 75° at time 0, which means that the samples can be considered hydrophobic. Nevertheless, a decrease in the water contact angle is observed to θ_w_ = 55° after 30 min. This effect is more pronounced in the composites, as it can be observed that the bioPP hardly changes the deposition angle during the characterization time. Essaby et al. [40] reported a similar effect for composites, where a significant decrease in the contact angle is measured during the first minutes of deposition but a small change is observed after about 10 min. As they also propose, the incorporation of a coupling agent is beneficial from the point of view of the hydrophobic behavior since the reduction of the gaps between the filler and the polymer matrix increases the water affinity and consequently increases the water contact angle.

## 4. Conclusions

The methodology proposed in this work allows the proper obtaining of wood plastic composites made of bioPP/MAS and PP-g-MA as a compatibilizer. With the twin screw extrusion process followed by an injection molding process, it is possible to obtain a proper dispersion of the particles as observed in the morphological study. With this methodology, a reinforcing effect is obtained in the samples with increased stiffness and tensile strength. In return, a reduction in the ductile properties is obtained by reducing the chain mobility with the addition of MAS. This reduced ductility is observed in the surface morphology, with lower plastic deformation marks. From a thermal point of view, DSC shows that the introduction of MAS has little effect on the thermal transitions. In TGA, a reduction in thermal stability is observed since the filler contains cellulose, hemicellulose, and lignin that start to degrade at low temperatures. This increase in stiffness is also observed in the thermomechanical analysis with an increase in the storage modulus of the samples. In addition, XRD tests show that the bioPP forms a α structure during crystallization. The addition of argan byproducts modified other properties, such as the water affinity. Proportionally with the filler content, there is an increase in the water uptake values and a reduction in the water contact angle measured due to the presence of hydroxyl groups with high affinity to water. Finally, the introduction of the filler leads to a modification of the color of the samples, making them similar to wood, so that the prepared composites can be used for the manufacture of products typically made of wood.

## Figures and Tables

**Figure 1 polymers-15-02743-f001:**
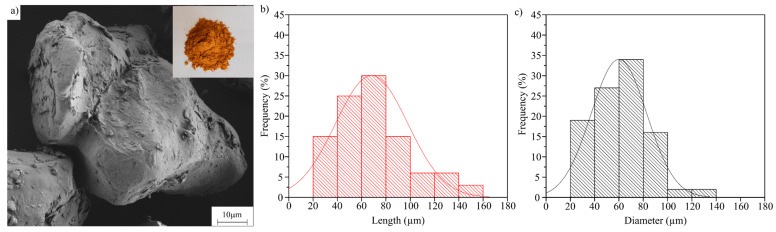
(**a**) Micronized Argan Shell (MAS) particles; (**b**) histogram of the MAS particle length; and (**c**) histogram of the MAS particle diameter.

**Figure 2 polymers-15-02743-f002:**
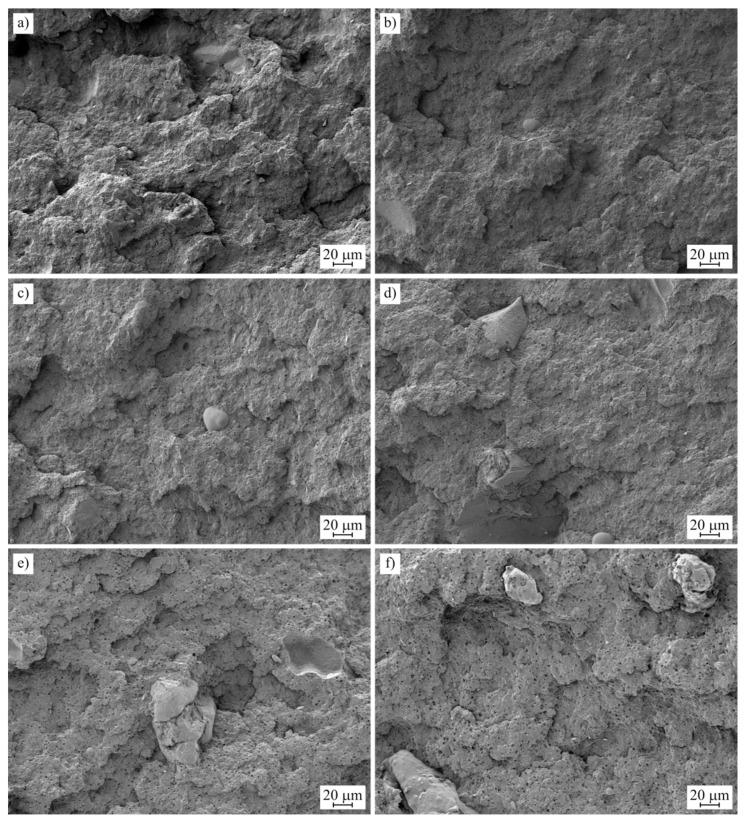
FESEM images at 250× of the fractured surfaces of: (**a**) bioPP; (**b**) bioPP/2.5MAS; (**c**) bioPP/5MAS; (**d**) bioPP/10MAS; (**e**) bioPP/20MAS; (**f**) bioPP/40MAS.

**Figure 3 polymers-15-02743-f003:**
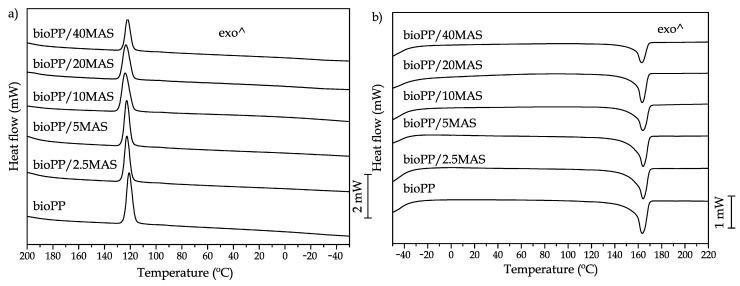
Main thermal transitions measured by DSC are (**a**) the cooling cycle and (**b**) the second heating.

**Figure 4 polymers-15-02743-f004:**
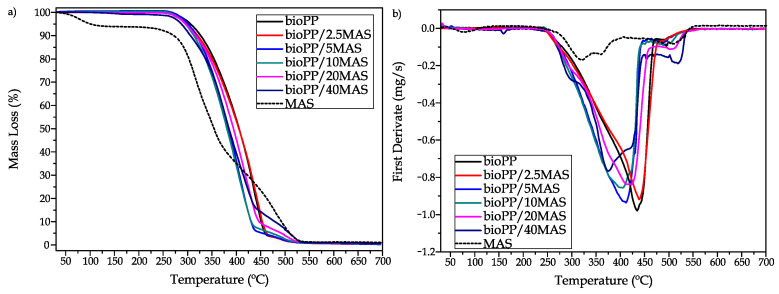
Thermal degradation of bioPP green composites in terms of: (**a**) mass loss and (**b**) first derivative.

**Figure 5 polymers-15-02743-f005:**
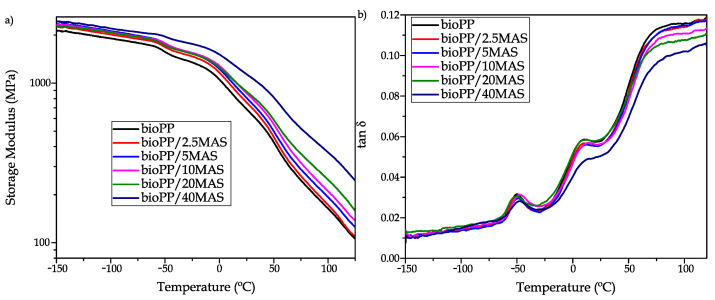
Evolution of: (**a**) storage modulus (E’) vs. temperature and (**b**) damping factor vs. temperature of bioPP composites.

**Figure 6 polymers-15-02743-f006:**
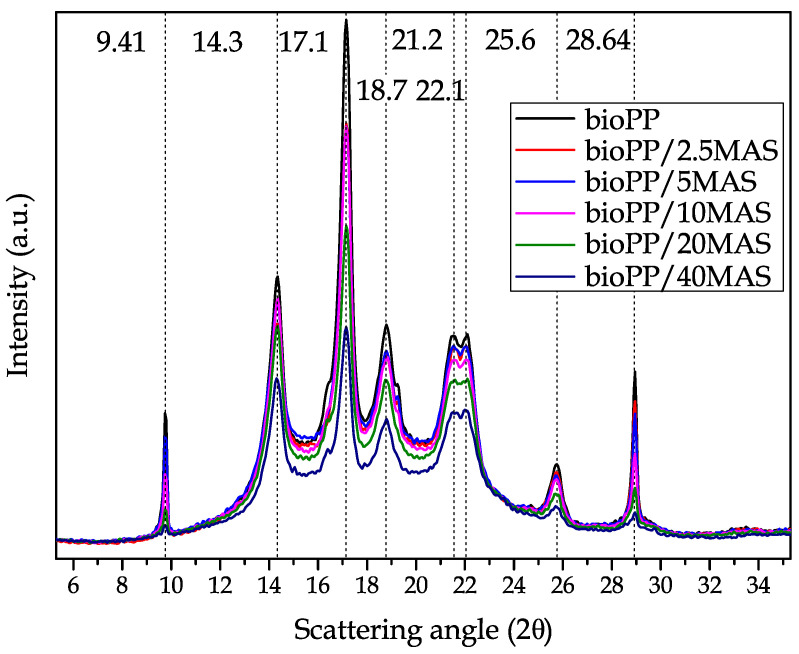
XRD comparative scans of the bioPP composite samples.

**Figure 7 polymers-15-02743-f007:**
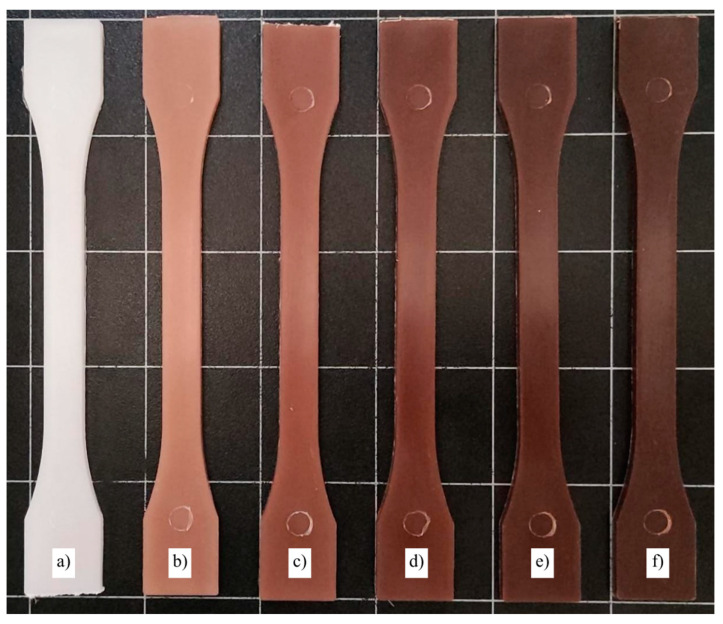
Visual appearance of the samples: (**a**) bioPP; (**b**) bioPP/2.5MAS; (**c**) bioPP/5MAS; (**d**) bioPP/10MAS; (**e**) bioPP/20MAS; (**f**) bioPP/40MAS.

**Figure 8 polymers-15-02743-f008:**
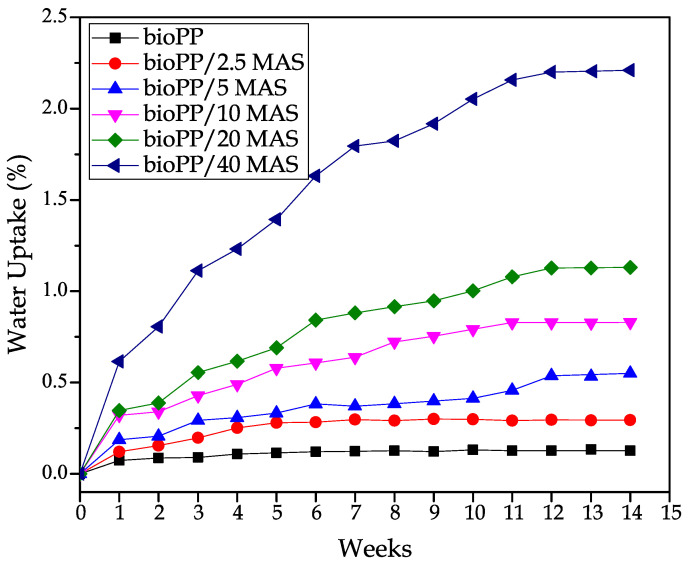
Water uptake of the bioPP/MAS composites as a function of the immersion time.

**Figure 9 polymers-15-02743-f009:**
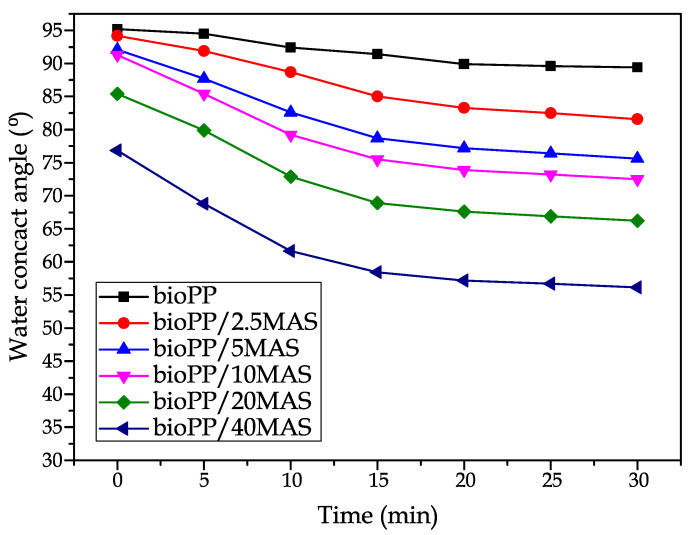
Water contact angle (θ_w_) of the samples at different times.

**Table 1 polymers-15-02743-t001:** Summary of the compositions of the green composites according to the amount of MAS and PP-g-MA.

Material	bioPP (%wt.)	MAS (%wt.)	PP-g-MA (phr)
bioPP	100	0	0
bioPP/2.5MAS	97.5	2.5	0.25
bioPP/5MAS	95	5	0.5
bioPP/10MAS	90	10	1.0
bioPP/20MAS	80	20	2.0
bioPP/40MAS	60	40	4.0

**Table 2 polymers-15-02743-t002:** Main mechanical properties of the bioPP composites: tensile modulus (E), maximum tensile strength (σ_max_), elongation at break (ε_b_), shore D hardness, and impact strength.

Code	E (MPa)	σ_max_ (MPa)	ε_b_ (%)	Shore D Hardness	Impact Strength (kJ/m^2^)
bioPP	1925 ± 91 ^a^	18.2 ± 0.3 ^a^	48.3 ± 1.4 ^a^	65.4 ± 2.1 ^a^	8.1 ± 0.4 ^a^
bioPP/2.5MAS	2029 ± 110 ^a^	18.9 ± 0.2 ^a^	24.9 ± 1.5 ^b^	66.7 ± 1.2 ^a^	5.0 ± 0.5 ^b^
bioPP/5MAS	2177 ± 105 ^b^	19.0 ± 0.5 ^a^	23.1 ± 0.7 ^b^	66.5 ± 1.3 ^a^	4.9 ± 0.4 ^b^
bioPP/10MAS	2237 ± 94 ^b^	19.7 ± 0.2 ^a^	18.9 ± 1.3 ^c^	67.0 ± 1.5 ^a^	4.7 ± 0.4 ^b^
bioPP/20MAS	2341 ± 127 ^b^	20.0 ± 0.2 ^a^	15.5 ± 1.6 ^c^	67.2 ± 1.4 ^a^	3.7 ± 0.3 ^c^
bioPP/40MAS	2435 ± 155 ^b^	20.8 ± 0.6 ^a^	9.9 ± 0.7 ^d^	70.0 ± 0.7 ^b^	1.8 ± 0.2 ^d^

^a–d^ Different letters in each column indicate a significant difference between the results (*p* < 0.05).

**Table 3 polymers-15-02743-t003:** Main thermal properties in terms of melt crystallization temperature (T_cm_), melt crystallization enthalpy (∆H_cm_), melting temperature (T_m_), melting enthalpy (∆*H_m_*), and degree of crystallinity (*X_c_*).

Code	T_cm_ (°C)	∆H_cm_ (J·g^−1^)	T_m_ (°C)	∆*H_m_* (J·g^−1^)	*X_c_* (%)
bioPP	120.9 ± 0.4 ^a^	74.6 ± 2.3 ^a^	163.5 ± 0.5 ^a^	67.0 ± 2.0 ^a^	32.1 ± 1.0 ^a^
bioPP/2.5MAS	122.4 ± 0.5 ^a^	72.0 ± 2.4 ^a^	164.5 ± 0.4 ^a^	63.5 ± 1.8 ^b^	31.2 ± 0.9 ^a^
bioPP/5MAS	123.9 ± 0.3 ^a^	70.5 ± 2.6 ^a^	164.2 ± 0.6 ^a^	61.5 ± 1.6 ^b^	31.0 ± 0.8 ^a^
bioPP/10MAS	122.5 ± 0.5 ^a^	64.5 ± 2.5 ^b^	163.9 ± 0.5 ^a^	56.3 ± 1.7 ^c^	29.9 ± 0.9 ^a^
bioPP/20MAS	122.6 ± 0.4 ^a^	62.5 ± 1.9 ^b^	163.4 ± 0.7 ^a^	50.1 ± 1.6 ^d^	30.0 ± 1.0 ^a^
bioPP/40MAS	121.4 ± 0.2 ^a^	45.5 ± 1.7 ^c^	163.0 ± 0.8 ^a^	37.2 ± 1.7 ^e^	29.7 ± 1.4 ^a^

^a–e^ Different letters in each column indicate a significant difference between the results (*p* < 0.05).

**Table 4 polymers-15-02743-t004:** Summary of the thermal degradation parameters of the bioPP/MAS composites in terms of the starting degradation temperature (*T*_5%_), maximum degradation rate temperature (*T_deg_*), and residual mass at 700 °C.

Code	*T*_5%_ (°C)	*T_deg_* (°C)	Residual Weight (%)
bioPP	305.3 ± 1.3 ^a^	435.0 ± 2.8 ^a^	0.2 ± 0.1 ^a^
bioPP/2.5MAS	299.5 ± 1.2 ^b^	438.7 ± 2.2 ^a^	0.3 ± 0.1 ^a^
bioPP/5MAS	297.5 ± 1.7 ^b^	411.1 ± 1.9 ^b^	0.3 ± 0.1 ^a^
bioPP/10MAS	293.8 ± 1.5 ^b^	403.2 ± 1.5 ^b^	0.6 ± 0.2 ^b^
bioPP/20MAS	294.2 ± 1.0 ^b^	416.1 ± 3.6 ^b^	0.7 ± 0.2 ^b^
bioPP/40MAS	287.9 ± 1.1 ^c^	374.6 ± 2.1 ^c^	0.8 ± 0.1 ^b^
MAS	110.5 ± 1.2 ^d^	321.8 ± 3.1 ^d^	0.6 ± 0.1 ^b^

^a–d^ Different letters in each column indicate a significant difference between the results (*p* < 0.05).

**Table 5 polymers-15-02743-t005:** Summary of the dynamic-mechanical properties of bioPP/MAS composites at different temperatures.

Code	E′ (MPa) at −125 °C	E′ (MPa) at 25 °C	E′ (MPa) at 100 °C	*T_g_* (°C) *
bioPP	2032 ± 45 ^a^	660 ± 19 ^a^	165 ± 5 ^a^	10.7 ± 0.4 ^a^
bioPP/2.5MAS	2150 ± 35 ^a^	740 ± 15 ^b^	175 ± 4 ^a^	10.2 ± 0.3 ^a^
bioPP/5MAS	2280 ± 45 ^a^	790 ± 18 ^b^	200 ± 8 ^b^	12.3 ± 0.1 ^b^
bioPP/10MAS	2240 ± 61 ^a^	850 ± 20 ^b^	210 ± 3 ^b^	14.9 ± 0.2 ^c^
bioPP/20MAS	2260 ± 50 ^a^	870 ± 15 ^b^	250 ± 2 ^c^	14.5 ± 0.3 ^c^
bioPP/40MAS	2375 ± 35 ^a^	1120 ± 30 ^c^	370 ± 6 ^d^	16.6 ± 0.4 ^d^

* The *T_g_* from the tan δ peak maximum criterion. ^a–d^ Different letters in each column indicate a significant difference between the results (*p* < 0.05).

**Table 6 polymers-15-02743-t006:** Color coordinates (*L*a*b**) and luminance of the bioPP green composites.

Code	*L**	*a**	*b**	${∆E}_{ab}^{*}$
bioPP	62.7 ± 0.1 ^a^	−1.77 ± 0.03 ^a^	−3.50 ± 0.06 ^a^	-
bioPP/2.5MAS	43.6 ± 0.6 ^b^	6.51 ± 0.14 ^b^	8.64 ± 0.22 ^b^	24.4 ^a^
bioPP/5MAS	36.4 ± 0.1 ^c^	7.82 ± 0.09 ^c^	8.67 ± 0.11 ^b^	30.5 ^b^
bioPP/10MAS	31.1 ± 0.1 ^d^	6.88 ± 0.11^d^	6.64 ± 0.13 ^c^	34.3 ^c^
bioPP/20MAS	28.2 ± 0.2 ^e^	6.13 ± 0.34 ^d^	6.14 ± 0.63 ^c^	36.7 ^d^
bioPP/40MAS	27.0 ± 0.1 ^e^	5.37 ± 0.15 ^e^	4.85 ± 0.12 ^d^	37.4 ^d^

^a–e^ Different letters in each column indicate a significant difference between the results (*p* < 0.05).

## Data Availability

Not applicable.

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
