# Peer review of "Incorporation of Argan Shell Flour in a Biobased Polypropylene Matrix for the Development of High Environmentally Friendly Composites by Injection Molding"

_polymers, 2023, doi:10.3390/polym15122743_

Round 1
Reviewer 1 Report
Dear Authors, below you can find my suggeston after reading of the manuscript.
Comments and Suggestions for Authors,
Comments and suggestions for authors, Manuscript titled: Incoporation of Argan Shell Flour in a Biobased Polypropylene Matrix for the Development of High Environmentally Friendly Composites by Injection Molding. Overall, the organization and structure of this review was clear. There are a few considerations that should be considered before publication. Section 2.1. In the experiment, the authors add micronized argan husk waste to polypropylene (bioPP). I would like to add information to the manuscript regarding the content of lignin and cellulose in micronized argan coatings. In the further part of the research, information on the content of pectins was also introduced, please also provide the content of pectins in the lignocellulosic material. The essence of the compatibility of bioPP and MAS is not only the size of the particles, but also their composition. The authors rightly noticed the lack of chemical compatibility between the components used in the production of the biocomposite. No less differently it will bind to bio-PP, to the polymer which is lignin, and differently to cellulose, which is a polysaccharide. How was the size and length of MAS particles determined? I mean the question about the methodology for determining the length and diameter. Particle sizes have a huge impact on the formation of intermolecular bonds, which is why this part of the methodology lacks information about the shape of the particles, and more specifically about the area of their development. I would like the authors to complete the information and add to this part of the manuscript microscopic photos of the MAS particle or a sketch showing the shape of the MAS particle. Section 2.2: What was the final moisture content of the bio-PP+MAS PP-g-MA mixture after the drying process? Section 2.3.3.: Were the DSC measurements carried out according to your own methodology or did you use a standardized methodology described in the ISO standard (e.g. ISO 11357-1:2016)? Section 2.3.6.: Was the instrument calibrated on a white standard before the measurements? Please provide such information. Was the color measurement repeated, how many times? Chapter Results: Describing the results presented in Table 2, they suggest: “….. this increase is to be expected because the filler has a higher hardness and this effect is also related to the good adhesion of the filler.” Does the comment concern the adhesion of the putty to bioPP or to PP-gMA? In my opinion, this is a more complex interaction phenomenon at the molecular level. Which, I think is nice shown in Figure 4b.
Author Response
Dear reviewer
We want to thank the efforts made in the review process carried out. In addition to your comments, the recommendations provided by the other reviewers have been implemented so that this manuscript can considered for its publication.
Comments and suggestions for authors,
Manuscript titled: Incoporation of Argan Shell Flour in a Biobased Polypropylene Matrix for the Development of High Environmentally Friendly Composites by Injection Molding. Overall, the organization and structure of this review was clear. There are a few considerations that should be considered before publication.
Section 2.1. In the experiment, the authors add micronized argan husk waste to polypropylene (bioPP). I would like to add information to the manuscript regarding the content of lignin and cellulose in micronized argan coatings. In the further part of the research, information on the content of pectins was also introduced, please also provide the content of pectins in the lignocellulosic material.
Answer
Thank you for this suggestion provided. In the revised version of the manuscript some information related with the composition of the filler has been included. In this case, the amount of pectin is very low as suggested by the manuscript cited.
“Regarding the shell composition, as it is proposed above lignocellulosic fillers are mainly composed by cellulose, hemicellulose and lignin. For the argan nut shells cellulose represents 48%, while lignin is 30% and hemicellulose is 16%. The remaining part of the composition has been reported as 6% ash, in which there is a small amount of pectin. [41].”
The essence of the compatibility of bioPP and MAS is not only the size of the particles, but also their composition. The authors rightly noticed the lack of chemical compatibility between the components used in the production of the biocomposite. No less differently it will bind to bio-PP, to the polymer which is lignin, and differently to cellulose, which is a polysaccharide. How was the size and length of MAS particles determined? I mean the question about the methodology for determining the length and diameter. Particle sizes have a huge impact on the formation of intermolecular bonds, which is why this part of the methodology lacks information about the shape of the particles, and more specifically about the area of their development. I would like the authors to complete the information and add to this part of the manuscript microscopic photos of the MAS particle or a sketch showing the shape of the MAS particle.
Answer
We highly appreciate this comment to improve the quality of the manuscript. In the revised version, the process followed for the particle observation has been included in the experimental section. As also suggested, an image with the particle morphology has been included. In addition new comments related with the effect of the particle size an surface morphology and its implication in the composite behaviour have been included.
“For MAS particle observation, the same conditions were employed. For the size characterization of the particles, ImageJ software was employed for the measurements (https://imagej.net).”
“The analysis of the filler morphology allows to understand the mechanical behavior of the composites obtained. With the proposed milling process the particles have a low surface roughness that limits the interaction with the polymer [50]. Another parameter that changes the mechanical behaviour of the composite is the particle size. In this case MAS filler particles have a small size is around 80 m. Essabir et al. analyzed the effect of the size of MAS particles in a PP matrix, as a result the best strengths were obtained for the composites prepare with the smaller particles that al-low to obtain a higher filler interaction with the polymer matrix [51].”
Section 2.2: What was the final moisture content of the bio-PP+MAS PP-g-MA mixture after the drying process?
Answer
Thank you for this suggestion. We do not control moisture content in the process as PP has no huge problems with moisture like the hydrolysis that occurs in PLA. The only problem that can occur is the void formation during the manufacture of the samples. Based on our experience in polymer composites, the drying process followed is enough to avoid this kind of problems.
“Drying process is an important parameter to consider before the manufacture of the samples since it can promote defects formation as holes. Nevertheless, drying process is not a critical parameter in polyolefins as hydrolytic degradation does not occur during the manufacture as can happen y polymer matrix as PLA [43].”
Section 2.3.3.: Were the DSC measurements carried out according to your own methodology or did you use a standardized methodology described in the ISO standard (e.g. ISO 11357-1:2016)?
Answer
During the DSC testing process, we employ our own method. This method was recommended by the equipment provider. We know that under this conditions proposed the results obtained are in concordance with the one obtained by other authors.
“Our own method was employed with an average weight of 6–7 mg of each material was subjected to a thermal program divided into three stages: a first heating from 30 °C to 200 °C followed by a cooling to -50 °C, and a second heating to 220 °C. Both heating and cooling rates were set to 10 °C/min.”
Section 2.3.6.: Was the instrument calibrated on a white standard before the measurements? Please provide such information. Was the color measurement repeated, how many times?
Answer
Thank you for this suggestion, the equipment employed always asks for a calibration after each initiation. For the calibration process, the methodology provided by the provider was followed. In addition, for the color measurements were repeated to obtain an average and a deviation.
Before the measurement, the equipment was calibrated following the procedure proposed by the provider. For each value provided, the measurements were performed five times in different tensile test samples.
Chapter Results: Describing the results presented in Table 2, they suggest: “….. this increase is to be expected because the filler has a higher hardness and this effect is also related to the good adhesion of the filler.” Does the comment concern the adhesion of the putty to bioPP or to PP-gMA? In my opinion, this is a more complex interaction phenomenon at the molecular level. Which, I think is nice shown in Figure 4b.
Answer
In order to satisfy the proposal made by the reviewer, some extra comments related with the hardness of the composites and the contribution of the filler to its increase have been included.
As they propose, this increase is expected since the filler has a higher hardness and also showed a good adhesion. Other way to increase the hardness in polymer composites is to introduce a nanoclay to enhance the filler interaction with the polymer [50]. Both examples demonstrate the importance of a good filler adhesion to improve the composite hardness.

Reviewer 2 Report
1. Please clarify how the compatibilizer content determined in this manuscript? Have the same MAS and different PP-g-MA samples been compared?
2. Can MAS be used anα nucleation agent?
3. Does the decrease of PP crystallinity affect the mechanical properties of bio PP/MAS?
4. It's recommended to supplement some bio PP/MAS water absorption data before adding compatibilizer.
There are few grammatical and typographical errors throughout the manuscript. Please revise the whole manuscript carefully.
Author Response
Dear reviewer
We want to thank the efforts made in the review process carried out. Apart from the comments, the recommendations related with the introduction, references, conclusion and so on have been included in the revised version of the manuscript.
In addition, the recommendations provided by the other reviewers have been implemented so that this manuscript can considered for its publication.
- Please clarify how the compatibilizer content determined in this manuscript? Have the same MAS and different PP-g-MA samples been compared?
Answer
The employment of a copolymer as PP-g-MA has been widely employed in many works as it is a simple strategy to improve the interaction of lignocellulosic filler into PP. In this case we employed PE-g-MA in a previous work with argan shells and we could assess that the optimum proportion of the copolymer is 1 part of copolymer for 10 parts of filler. In order make it clearer, some extra sentences have been included in the manuscript.
“For the PP-g-MA content in each sample, a proportion of 1 to 10 from the MAS content in phr has been considered. This proportion was selected according to a previous work in which MAS was successfully compatibilized into a HDPE formulation [42].”
- Can MAS be used anα nucleation agent?
Answer
Thank you for this interesting suggestion. As you suggest, the addition of MAS provides a change in terms of the degree of crystallinity so it can be expected that it could used as a nucleating agent. In literature, nucleating agents are added in small amounts and provide high changes in terms of degree of crystallinity. In this case the changes in this property are small and the amount of filler included is quite big.
“Typical nucleating agents are included in small quantities and induce a high change in terms of degree of crystallinity in the sample [56].”
- Does the decrease of PP crystallinity affect the mechanical properties of bio PP/MAS?
Answer
We highly appreciate this comment from the reviewer. In order to answer this suggestion, some extra references in which the effect of the degree of crystallinity on the properties of the samples is discussed.
“Some studies related with the nucleation effects in a PP matrix have reported that with the modification of the typical α structure to a β structure promotes changes in mechanical properties. For examples is known that β structure enhances the energy dissipation allowing to improve the toughness of the samples [58]. The degree of crystallinity has also an effect on a PP manufactured by injection molding with different material molds in terms of mechanical properties induced by the modification of the cooling conditions. In this regard an aluminum mold promotes a higher cooling rate and a result a lower degree of crystallinity compared to a polyacrylonitrile-butadiene-styrene (ABS) mold. With the lower degree of crystallinity an improvement of the ductile properties could be observed [59].”
- It's recommended to supplement some bio PP/MAS water absorption data before adding compatibilizer.
Answer
Thank you for this recommendation. We do not have a sample without the compatibilizer, so we cannot provide this information. To perform this test, we need a huge amount of time to provide relevant data. Instead of providing information we decided to include an extra reference in which the compatibilizers are successfully employed to modify the
Different strategies have been successfully employed reduce the water uptake of the polymer composites made with lignocellulosic fillers, all of them try to reduce the voids between the filler the polymer, reduce the hydroxyl groups exposed in the filler surface or both of the effects at the same time [79].

Reviewer 3 Report
The manuscript “Incoporation of Argan Shell Flour in a Biobased Polypropylene Matrix for the Development of High Environmentally Friendly Composites by Injection Moulding” investigates the possibility of reinforcing PP from renewable sources with Argan shell byproducts. The study considers the effect of adding this filler along with a compatibilizer on the structure and thermal and mechanical properties of the resulting composite.
The work is well written and clear, here I report some questions that should be addressed:
Sample preparation
The authors here report the components mixing procedure and composite injection molding procedure.
The mixing procedure is essential to matrix compatibilization and appropriate filler distribution and dispersion, but can also cause a partial degradation of the matrix (which may be a co-cause of composite embrittlement). Was this point considered, maybe extruding also the unfilled material? If it is not the case, how was the point assessed?
In addition to melt temperatures and fill time, would it be possible to indicate the packing pressure, mold temperature and mold geometry?
In table A, it should be indicated if the relative % of components are reported on a mass or volume basis.
Results
Mechanical properties
The moduli of both bioPP and its composites are surprisingly low: from my experience, and a fast survey of PP datasheets (and campusplastics.com), PP shows a tensile modulus on the order of magnitude of 1GPa, while in this case it is five times lower. Also the data in DTMA in a later section in the manuscript suggest that the modulus is higher than what reported here: how was the stress and, more importantly, the strain measured? I suggest checking the reported data. Possibly, some stress-strain curves might be presented, if not in the manuscript at least as supplementary information.
Independent of the method applied to estimate tensile modulus, a clear effect of the reinforcing agent is observed. It would be interesting to check if some model from literature (from the simplest rule of mixtures to more sophisticated Halpin-Tsai or similar micromechanical models) apply to the data.
Finally for this section, authors report an improvement in the maximum tensile strength (lines 225-227), however in table 2, if I correctly get the meaning of a-d superscripts, the increase in strength is never statistically significant, so the analysis of data seems to be contradictory with respect to the text. Or is not in table 2 misleading?
What instead is clear is evident is the reduction in deformation at break and in impact strength: the authors attribute this to a limitation of the bioPP mobility to the presence of the filler. May there also be an effect of the defects induced by the rigid particle into the matrix (despite the compatibilization)?
Thermal properties
The authors discuss (lines 297 and following) the effect of adding MAS on crystallization and on crystallinity. Table 3 reports the results: why the enthalpy of the peak measured during cooling is systematically higher than that during the following heating? What contributes to crystallization, but then does not melt on heating?
Figure 4 reports TGA plots for the considered composites: would it be possible to add the TGA plots for MAS alone?
Besides these questions, below I suggest some minor corrections for the text
Title: Amend INCOPORATION in INCORPORATION
Line 202: Consider changing “Distilled water drops were deposited at random in the surface of impact test samples” with “Distilled water drops were deposited at random positions in the surface of impact test samples”
Line 214: Consider changing “strength” with “tests” ( “strength tests”)
Line 303: Check the sentence “of the here in prepared composites.”
Line 381: Consider changing “recrystallize” with “crystallize”
Line 389: Amend “graffite” in “graphite”
Line 451-452: Check the sentence: “In addition, with the introduction of a filler increases the water uptake is increased.”
Line 468: Amend “a void” in “avoid”
Line 469: consider changing “this gaps are responsible to increase” into “being these gaps responsible for increasing ”
Line 509: consider changing “This” in “The”
Minor text corrections are in the comments to author section
Author Response
Dear reviewer
We want to thank the efforts made in the review process carried out. In addition to your comments, the recommendations provided by the other reviewers have been implemented so that this manuscript can considered for its publication.
The manuscript “Incoporation of Argan Shell Flour in a Biobased Polypropylene Matrix for the Development of High Environmentally Friendly Composites by Injection Moulding” investigates the possibility of reinforcing PP from renewable sources with Argan shell byproducts. The study considers the effect of adding this filler along with a compatibilizer on the structure and thermal and mechanical properties of the resulting composite.
The work is well written and clear, here I report some questions that should be addressed:
Sample preparation
The authors here report the components mixing procedure and composite injection molding procedure. The mixing procedure is essential to matrix compatibilization and appropriate filler distribution and dispersion, but can also cause a partial degradation of the matrix (which may be a co-cause of composite embrittlement). Was this point considered, maybe extruding also the unfilled material? If it is not the case, how was the point assessed?
Answer
Authors want to tank this comment. In our works we always apply the same thermal cycle to all the materials. We are aware that the heating process over the melting temperature can induce some degradation on the materials. By applying the same thermal cycle to all of them, this is not a parameter to consider during the analysis of the material properties. In addition we employed a temperature as low as possible to avoid the MAS degradation. The working temperatures are selected according to the matrix melting point, material has to be properly melted in order to achieve a good dispersion during the extrusion process.
“All materials were extruded under the same conditions, even neat polymer was extruded so the same thermal treatment is applied to all the samples. Temperatures were selected as low as possible, so MAS filler was not degraded during manufacturing.”
In addition to melt temperatures and fill time, would it be possible to indicate the packing pressure, mold temperature and mold geometry?
Answer
Thank you for this comment. In the new version of the manuscript this information has been provided.
“A pressure of 30 bar was maintained for 10 seconds to avoid any defects in the final samples. An aluminum mold was kept at 30 °C during all process and the geometry of the cavity produces tensile test samples according to ISO 527-1:2012 with samples 1B and rectangular samples with dimensions of 80 × 10 × 4 mm3”
In table A, it should be indicated if the relative % of components are reported on a mass or volume basis.
Answer
Thank you for this comment, in the revised version this change has been implemented.
Results
Mechanical properties
The moduli of both bioPP and its composites are surprisingly low: from my experience, and a fast survey of PP datasheets (and campusplastics.com), PP shows a tensile modulus on the order of magnitude of 1GPa, while in this case it is five times lower. Also the data in DTMA in a later section in the manuscript suggest that the modulus is higher than what reported here: how was the stress and, more importantly, the strain measured? I suggest checking the reported data. Possibly, some stress-strain curves might be presented, if not in the manuscript at least as supplementary information.
Answer
We highly appreciate this comment. It is true that the information provided in the first version of this manuscript is not completely correct. The previous data was collected without the employment of extensometer. In our universal testing machine is recommended to employe this device in order to get a good precision during the tensile modulus calculation.
In order to solve this, we have repeated the measurements and the new information is provided in the table below. In addition, the information related with the extensometer is included in the experimental section.
“For tensile modulus measurement, a 3542-050M-050-ST extensometer from Epsilon Technology Corporation (Jackson, WY, USA) was employed.”
Code E (MPa) σmax (MPa) εb (%) Shore D Hardness
bioPP 1925 ± 91a 18.2 ± 0.3a 48.3 ± 1.4a 65.4 ± 2.1a
bioPP/2.5MAS 2029 ± 110a 18.9 ± 0.2a 24.9 ± 1.5b 66.7 ± 1.2a
bioPP/5MAS 2177 ± 105b 19.0 ± 0.5a 23.1 ± 0.7b 66.5 ± 1.3a
bioPP/10MAS 2237 ± 94b 19.7 ± 0.2a 18.9 ± 1.3c 67.0 ± 1.5a
bioPP/20MAS 2341 ± 127b 20.0 ± 0.2a 15.5 ± 1.6c 67.2 ± 1.4a
bioPP/40MAS 2435 ± 155b 20.8 ± 0.6a 9.9 ± 0.7d 70.0 ± 0.7b
Independent of the method applied to estimate tensile modulus, a clear effect of the reinforcing agent is observed. It would be interesting to check if some model from literature (from the simplest rule of mixtures to more sophisticated Halpin-Tsai or similar micromechanical models) apply to the data.
Answer
We appreciate this comment from the reviewer. In this case we consider that the manuscript is enough long in this moment and new information would make it too long. For future works we would consider this suggestion to be implemented from the beginning.
Finally for this section, authors report an improvement in the maximum tensile strength (lines 225-227), however in table 2, if I correctly get the meaning of a-d superscripts, the increase in strength is never statistically significant, so the analysis of data seems to be contradictory with respect to the text. Or is not in table 2 misleading?
Answer
As the reviewer suggest, changes in terms of tensile strength do not provide statistical differences. In the new version of the manuscript the discussion in terms of tensile strength has been changed. Now it is highlighted that only small changes occur but are not enough big to provide significant differences.
“In this work, despite a small improvement in the tensile strength is obtained, the differences emerged do not provide a significant difference between the materials. Even the reinforcement effect expected by the MAS addition, the disruption provided by the particles do not allow to increase the tensile strength of the composites.”
What instead is clear is evident is the reduction in deformation at break and in impact strength: the authors attribute this to a limitation of the bioPP mobility to the presence of the filler. May there also be an effect of the defects induced by the rigid particle into the matrix (despite the compatibilization)?
Answer
As the reviewer suggest, a reduction in the chain mobility is not the only effect that induces the filler introduction. Other effects like the stress concentration due to the particle presence in the structure affect the ductile properties of the composite materials. To improve the discussion, also this effect has been commented as mechanism to reduce the mechanical properties.
“In addition the presence of a fillers induces a stress concentration effect that promotes the crack formation, as a result the energy absorption of the composites is reduced [52].”
Thermal properties
The authors discuss (lines 297 and following) the effect of adding MAS on crystallization and on crystallinity. Table 3 reports the results: why the enthalpy of the peak measured during cooling is systematically higher than that during the following heating? What contributes to crystallization, but then does not melt on heating?
Answer
Thank you for your huge effort to review this manuscript. In this case this effect has been also reported by other authors as can be observed in the new reference included. As discussed in the manuscript, the formation of the crystals requires more energy than the melting process.
“Another phenomenon to highlight related with the enthalpies is that the values measured during cooling are higher than in the heating. This phenomenon has been observed by other authors like Achaby et al. due to the crystal formation is more energy consuming than the melting process [61].”
Figure 4 reports TGA plots for the considered composites: would it be possible to add the TGA plots for MAS alone?
Answer
As the reviewer proposed, the TGA curves for MAS have been added. In addition some extra comments have been included to discuss the MAS degradation process.
Besides these questions, below I suggest some minor corrections for the text
Title: Amend INCOPORATION in INCORPORATION
Line 202: Consider changing “Distilled water drops were deposited at random in the surface of impact test samples” with “Distilled water drops were deposited at random positions in the surface of impact test samples”
Line 214: Consider changing “strength” with “tests” ( “strength tests”)
Line 303: Check the sentence “of the here in prepared composites.”
Line 381: Consider changing “recrystallize” with “crystallize”
Line 389: Amend “graffite” in “graphite”
Line 451-452: Check the sentence: “In addition, with the introduction of a filler increases the water uptake is increased.”
Line 468: Amend “a void” in “avoid”
Line 469: consider changing “this gaps are responsible to increase” into “being these gaps responsible for increasing ”
Line 509: consider changing “This” in “The”
Answer
We really want to thank the effort made by the reviewer work. The modifications suggested have been implemented in the revised version.

Round 2
Reviewer 1 Report
Dear authors, thank you for making corrections to the manuscript and taking my comments into account. In my opinion, the manuscript may be published in the journal polymers. Best regardsReviewer